# Hatsusamides A and B: Two New Metabolites Produced by the Deep-Sea-Derived Fungal Strain *Penicillium steckii* FKJ-0213

**DOI:** 10.3390/md18100513

**Published:** 2020-10-12

**Authors:** Hirotaka Matsuo, Rei Hokari, Aki Ishiyama, Masato Iwatsuki, Mayuka Higo, Kenichi Nonaka, Yuriko Nagano, Yōko Takahashi, Satoshi Ōmura, Takuji Nakashima

**Affiliations:** 1Ōmura Satoshi Memorial Institute, Kitasato University, 5-9-1 Shirokane, Minatok-ku, Tokyo 108-8641, Japan; hokari@lisci.kitasato-u.ac.jp (R.H.); ishiyama@lisci.kitasato-u.ac.jp (A.I.); iwatsuki@lisci.kitasato-u.ac.jp (M.I.); mayuka@lisci.kitasato-u.ac.jp (M.H.); ken@lisci.kitasato-u.ac.jp (K.N.); ytakaha@lisci.kitasato-u.ac.jp (Y.T.); omuras@insti.kitasato-u.ac.jp (S.Ō.); 2Department of Drug Discover Sciences, Graduate School of Infection Control Sciences, 5-9-1 Shirokane, Minatok-ku, Tokyo 108-8641, Japan; 3Research Center for Medicinal Plant Resources, National Institutes of Biomedical Innovation, Health and Nutrition, 1-2 Hachimandai, Tsukuba, Ibaraki 305-8043, Japan; 4Department of Marine Biodiversity Research, Japan Agency for Marine-Earth Science and Technology, 2-15 Natsushima-cho, Yokosuka, Kanagawa 237-0061, Japan; y.nagano@jamstec.go.jp; 5Research Innovation Center, Waseda University, 513 Waseda tsurumakicho, Shinjuku-ku, Tokyo 162-0041, Japan

**Keywords:** anti-malarial activity, anti-tumor activity, deep-sea-derived fungus, hatsusamides A and B, tanzawaic acid B, trichodermamide C

## Abstract

Two new nitrogen-containing metabolites, designated hatsusamide A (**1**) and B (**2**), were isolated from a culture broth of *Penicillium*
*steckii* FKJ-0213 together with the known compounds tanzawaic acid B (**3**) and trichodermamide C (**4**) by physicochemical (PC) screening. The structures of **1** and **2** were determined as a tanzawaic acid B-trichodermamide C hybrid structure and a new analog of aspergillazines, respectively. The absolute configuration of **1** was determined by comparing the values of tanzawaic acid B and trichodermamide C in the literatures, such as ^1^H-nuclear magnetic resonance (^1^H-NMR) data and optical rotation, after hydrolysis of **1**. Compounds **1**–**4** were evaluated for cytotoxicity and anti-malarial activities. Compounds **1** and **3** exhibited weak anti-malarial activity at half-maximal inhibitory concentration (IC_50_) values of 27.2 and 78.5 µM against the K1 strain, and 27.9 and 79.2 µM against the FCR3 strain of *Plasmodium falciparum*, respectively. Furthermore, **1** exhibited cytotoxicity against HeLa S3, A549, Panc1, HT29 and H1299 cells, with IC_50_ values of 15.0, 13.7, 12.9, 6.8, and 18.7 μM, respectively.

## 1. Introduction

Fungal metabolites have been utilized as drugs and as agricultural and chemical reagents. Fungal metabolites containing nitrogen atoms are especially important due to their strong and unique activities. For example, oxaline and neoxaline, discovered from the culture broth of *Aspergillus japonicus* Fg-551, exhibit mitotic activity and have been used to guide the development of anti-mitotic agents [1,2,3,4]. Moreover, in the Kyoto Encyclopedia of Genes and Genomes (KEGG) MEDICUS (https://www.kegg.jp/kegg/medicus/) database, which presents information on commercially available medicines [5], 87% of the listed medicines contain nitrogen atoms. Thus, nitrogen-containing metabolites are expected to be widely used as a source of medicines.

Marine-derived fungi have been an important source for the discovery of structurally and pharmaceutically useful new compounds [6]. In particular, the search for novel and bioactive compounds for drug discovery from deep-sea-derived fungi has become an increasingly important area of research [7]. The deep-sea is generally defined as the region 200 m below the sea surface and is characterized by extreme environmental conditions, such as darkness, low temperature, and high pressure. Nonetheless, recent research has revealed that fungi are abundant in the deep sea [8]. Our groups have isolated many fungal strains from deep-sea environments and used them as sources for new chemical compounds. For example, sarcopodinols A and B were discovered from the culture broth of the deep-sea-derived fungal strain *Sarcopodium* sp. FKJ-0025. After the discovery of sarcopodinols A and B, these compounds were found to exhibit anti-tumor activity [9]. Cipralphelin was discovered from the culture broth of the deep-sea-derived fungal strain *Penicillium brevicompactum* FKJ-0123 [10] and shown to exhibit hydroxy radical-scavenging activity.

One useful method for the discovery of new compounds is the physicochemical (PC) screening method, which uses an index of physicochemical properties, such as molecular weight, molecular formula, ultraviolet (UV) spectrum, and chemical characteristics to identify compounds [11]. We have used PC screening in combination with an in-house database and natural product databases, such as the Dictionary of Natural Products (http://dnp.chemnetbase.com/), to discover new nitrogen-containing compounds from deep-sea-derived fungi. During our recent PC screening, two fungal metabolites, which had a mass-to-charge ratio (*m*/*z*) of 703.3221 [M + H]^+^ and 449.1560 [M + H]^+^, were determined to be new nitrogen-containing metabolites by analyzing molecular formula and Dragendorff’s reagent. The producing strain FKJ-0213 was isolated from a sediment sample collected at a depth of 1171 m off the island of Hatsushima (Shizuoka Prefecture, Japan) and identified as *Penicillium steckii* by DNA barcoding. As a result of purification guided by LC/MS analyses from the culture broth of fungal strain FKJ-0213, the new metabolites, designated hatsusamide A (**1**) and B (**2**), were isolated together with two known compounds, tanzawaic acid B (**3**) and trichodermamide C (**4**) [12,13,14]. Interestingly, structure elucidation revealed that **1** had a tanzawaic acid B-trichodermamide C hybrid structure. Here, we report the fermentation, isolation, structure elucidation, and some biological activity of **1**–**4**.

## 2. Results and Discussion

### 2.1. Structure Elucidation of ***1*** and ***2***

Compound **1** was isolated as a yellow amorphous solid and determined to have the molecular formula C_39_H_46_N_2_O_10_ by high-resolution electrospray ionization (HR-ESIMS) (*m*/*z* 703.3221 [M + H]^+^, calcd. for 703.3225). The ^1^H NMR (nuclear magnetic resonance) (Appendix A and Table 1) and HMQC (Heteronuclear Multiple Quantum Coherence) spectra in (CD_3_)_2_CO showed the presence of two aromatic protons, nine olefinic protons, three oxymethine protons, two methoxy protons, an *N*-methyl proton, and three methyl protons. The ^13^C (Appendix A and Table 1) NMR spectrum showed the presence of four oxygenated carbons, seventeen unsaturated carbons (including five carbons seemingly adjacent to a hetero atom), and three carbonyl carbons. The gross structure of **1** was deduced from detailed analyses of 2D NMR data, including ^1^H-^1^H COSY (Correlation Spectroscopy), HMQC, and HMBC (Heteronuclear Multiple Bond Correlation) pectra in (CD_3_)_2_CO (Appendix A). The ^1^H–^1^H COSY spectra revealed the presence of three partial structures **a**, **b**, and **c**, as shown in Figure 1. The planar structure of **1** was deduced to have a tanzawaic acid B unit by the partial structure **a** and the HMBC cross-peaks of H-3″ and H-2″ to C-1″, H-8″ to C-10″, H-9″ to C-7″ and C-15″, H-10″ to C-8″, and H-15″ to C-9″. The partial structure **b** and HMBC cross-peaks of H-3 to C-1, C-2, C-5 and C-9, H-5 to C-3, C-7 and C-9, and H-6 to C-4 and C-8 revealed the presence of a coumarin unit. The positions of two methoxy groups were determined by the HMBC cross-peaks of C-7-methoxy protons to C-7 and C-8-methoxy protons to C-8. The HMBC cross-peaks of *N*-methyl protons to C-1′ and C-2 revealed the presence of an *N*-methyl amide group at C-2. The partial structure of **c** and HMBC cross-peaks of H-5′ to C-4′, C-6′ and C-7′, H-6′ to C-4′ and C-8′, H-7′ to C-′5 and C-9′, and H-9′ to C-4′ revealed the presence of 2-cyclohexene-1-ol. When the structure, including *N*-methyl amide coumarin and 2-cyclohexene-1-ol, was searched using the Dictionary of Natural Products database, it was found to correspond with trichodermamide C. As **1** was considered to consist of tanzawaic acid B ester-linked trichodermamide C, the structure of **1** was confirmed from hydrolysis products. Compound **1** was hydrolyzed by 1 M NaOH at room temperature for 2 h. After neutralization, the hydrolysate was analyzed by LC/MS (Liquid chromatography–mass spectrometry) (Appendix A) and purified by preparative HPLC (High Performance Liquid Chromatography). The two isolated products were identified as tanzawaic acid B and trichodermamide C by comparison with the ^1^H-NMR (Appendix A) and optical rotation values in the literature. Therefore, the structure of **1** was suggested to be a hybrid structure of tanzawaic acid B and trichodermamide C. Finally, the connectivity of these two known compounds was determined by the HMBC cross peak of H-5′ to C-1″.

The relative configuration of tanzawaic acid B and the absolute configuration of trichodermamide C have been reported previously [12,13,14]. To determine the absolute configuration of **1**, tanzawaic acid B (**3**) was crystalized to carry out a single a crystal X-ray structure analysis. As a result of the analysis, the absolute configuration of tanzawaic acid B was determined, as shown in Appendix A. Therefore, the absolute configuration of **1** was determined to be 4′*S*, 5′*R*, 8′*R*, 9′*S*, 6″*R*, 7″*R*, 10″*S*, 12″*R*, 14″*S,* and 15″*R* (Figure 2).

Compound **2** was isolated as a pale yellow amorphous solid and revealed to have the molecular formula C_21_H_24_N_2_O_9_ by HR-ESIMS (High-resolution Electronspray Ionization Mass Spectrometry) (*m*/*z* 449.1560 [M + H]^+^, calcd. for 449.1554). The ^1^H NMR (Appendix A and Table 1) and HMQC spectra in CD_3_OD showed the presence of two aromatic protons, three olefinic protons, two oxymethine protons, two methoxy protons, and an *N*-methyl proton. The ^13^C NMR (Appendix A and Table 1) spectrum showed the presence of four oxygenated carbons, ten unsaturated carbons (including three carbons seemingly adjacent to a hetero atom), and two carbonyl carbons. The gross structure of **2** was deduced from detailed analyses of 2D NMR data, including ^1^H-^1^H COSY, HMQC, and HMBC spectra in CD_3_OD (Appendix A). The ^1^H–^1^H COSY spectra revealed the presence of three partial structures **a**, **b**, and **c**, as shown in Figure 3A. The presence of 1,2,3-trihydroxy-4-cyclohexene was revealed by the partial structures **a** and **b** and the HMBC cross-peaks of H-5′ to C-4′ and C-7′, H-6′ to C-7′ and C-8′, H-7′ to C-5′ and C-9′, H-8′ to C-4′ and C-6′, and H-9′ to C-7′. The HMBC cross-peaks of H-3′ to C-2′, C-4′, and C-9′, H-5′ to C-3′, and H-9′ to C-2′, and C-4′ revealed the presence of a tetrahydrofuran unit connecting at C-4′ and C-9′ of the cyclohexene unit. The partial structure **c** and HMBC cross-peaks of H-3 to C-1, C-5, and C-9, H-5 to C-2 (^4^*J*), C-3, C-7, and C-9, and H-6 to C-4 and C-8 revealed the presence of a coumarin unit. The positions of two methoxy groups were determined by the HMBC cross-peaks of C-7-methoxy protons to C-7 and C-8-methoxy protons to C-8. The HMBC cross-peaks of *N*-methyl protons to C-1′ and C-2 revealed the presence of an *N*-methyl amide group at C-2. The gross structure was determined by the HMBC cross-peaks of H-3′ to C-1′. Finally, the position of an amino group was determined at C-2′ by a chemical shift. Thus, the planar structure of **2** was determined as a new *N*-methyl analog of aspergillazines D and E (Figure 3A) [15].

The partial relative configuration of **2** was deduced by the ROESY (rotating-frame nuclear Overhauser effect correlation spectroscopy) spectrum, coupling constant and putative biogenetic grounds. From the ROESY correlation between H-6′ (*δ*_H_ 4.14) and H-3′ (*δ*_H_ 3.04), these protons had a β-orientation (Figure 3B). The coupling constant between H-5′ (*δ*_H_ 3.70) and H-6′ was 8.0 Hz, indicating that the proton at C-5 has an α-orientation (Figure 3B). Aspergillazines D and E are reported to be biogenetically related to penicillazine and epimerize at C-2 position. The equilibrium ratio of C-2 epimers is 1:0.85 [15,16]. Detail analyses of 1D and 2D-NMR showed that **2** also epimerized at C-2′ position. However, the equilibrium ratio of C-2′ epimers was 1:0.13 (Appendix A). The only difference from aspergillazines D and E, the *N*-methyl group, is probably involved in the suppression of the epimerization at C-2′ position. Compound **2** is also considered to be produced in the process of producing penicillazine and aspergillazines. Thus, the partial relative configuration of **2** was deduced, as shown in Figure 3B.

### 2.2. Anti-Malarial Activity of ***1**–**4***

Compounds **1**–**4** were tested for anti-malarial activity against both a chloroquine-resistant K1 strain and chloroquine-sensitive FCR3 strain of *Plasmodium falciparum* (Table 2). Compounds **1** and **3** showed anti-malarial at half-maximal inhibitory concentration (IC_50_) values of 27.2 and 78.5 µM against the K1 strain and 27.9 and 79.2 µM against the FCR3 strain of *P. falciparum*, respectively. Compounds **2** and **4** were inactive against both strains. Chloroquine was used as a positive control.

### 2.3. Cytotoxicity of ***1**–**4*** against Five Human Cancer Cell Lines

Compounds **1**–**4** were evaluated for cytotoxicity against HeLa S3, A549, Panc1, HT29 and H1299 cells. Compound **1** showed cytotoxicity against HeLa S3, A549, Panc1, HT29 and H1299 cells, with IC_50_ values of 15.0, 13.7, 12.9, 6.8, and 18.7 μM, respectively (Table 3). Compounds **2**–**4** were inactive against these five cell lines. Staurosporine was used as a positive control.

## 3. Materials and Methods

### 3.1. General Experimental Procedures

All solvents were purchased from Kanto Chemical (Tokyo, Japan). Silica gel and octa-decanoyl-silicon (ODS) were purchased from Fuji Silysia Chemical (Aichi, Japan).

High-resolution electrospray ionization (HR-ESIMS) spectra were measured using an AB Sciex TripleTOF 5600+ System (AB Sciex, Framingham, MA, USA). Nuclear magnetic resonance (NMR) spectra were measured using a JEOL JNM-ECA 500 spectrometer (JEOL, Tokyo, Japan), with ^1^H-NMR at 500 MHz and ^13^C NMR at 100 MHz in chloroform-*d* (CDCl_3_), methanol-*d*_4_ (CD_3_OD), and acetone-*d*_6_ [(CD_3_)_2_CO]. The chemical shifts are expressed in parts per million (ppm) and are referenced to residual C*H*Cl_3_ (7.26 ppm), C*H*D_2_OD (3.31 ppm), and C*H*D_2_COCD_3_ (2.05 ppm) in the ^1^H-NMR spectra and CDCl_3_ (77.0 ppm), CD_3_OD (49.0 ppm), and (CD_3_)_2_CO (29.8 ppm) in the ^13^C-NMR spectra. Infrared radiation (IR) spectra (KBr) were taken on a JASCO FT/IR-4600 Fourier Transform Infrared Spectrometer (JASCO, Tokyo, Japan). UV spectra were measured with a Hitachi U-2810 spectrophotometer (Hitachi, Tokyo, Japan). Optical rotation was measured on a JASCO model DIP-1000 polarimeter (JASCO, Tokyo, Japan). X-ray crystal analysis was measured on a RASA-5R (Rigaku, Akishima, Japan).

### 3.2. Isolation and Identification of Strain FKI-0213

Fungal strain FKJ-0213 (JAMSTEC ID: NT10-19 S01-8) was isolated from a deep-sea sediment sample at the bottom of the Sagami Bay off Hatsushima island, Shizuoka, Japan. This strain was identified as a member of the genus *Penicillium* based on its verticillate conidiophores and conidial chains. The internal transcribed spacer (ITS) sequence of FKJ-0213 was compared with sequences in the GenBank database using BLASTN 2.8.1 analyses [18]. The sequence of FKJ-0213 was in complete agreement with the sequence of CBS 260.55 (ex-type of *Penicillium steckii*, GenBank accession number NR_111488). Therefore, the producing strain FKJ-0213 was identified as *Penicillium steckii* based on its morphology and DNA barcoding.

### 3.3. Fermentation of Strain FKJ-0213 and Isolation of ***1**–**4***

Strain FKJ-0213 was grown on a modified Miura’s medium (LcA: consisting of 0.1% glycerol, 0.08% KH_2_PO_4_, 0.02% K_2_HPO_4_, 0.02% MgSO_4_·7H_2_O, 0.02% KCl, 0.2% NaNO_3_, 0.02% yeast extract, and 1.5% agar (adjusted to pH 6.0 before sterilization)) slant. A loop of spores of the strain was inoculated into a 500-mL Erlenmeyer flask containing 100 mL seed medium consisting of 2% glucose, 0.2% yeast extract, 0.5% hipolypeptone, 0.1% KH_2_PO_4_, 0.05% MgSO_4_·7H_2_O, and 0.1% agar, which was shaken at 210 rpm on a rotary shaker at 27 °C for 3 days. A 1-mL portion of the seed culture was transferred into 500-mL Erlenmeyer flasks (total 15 flasks) containing 100 mL of medium (3% soluble starch, 1% glycerol, 2% soybean meal, 0.3% dry yeast, 0.3% KCl, 0.2% CaCO_3_, 0.05% KH_2_PO_4_, 0.05% MgSO_4_·7H_2_O), and shaken at 210 rpm on a rotary shaker at 27 °C for 6 days.

The 6-day-old culture broth was supplemented with an equal volume of EtOH. The EtOH broth was filtrated and evaporated EtOH. The aqueous layer was partitioned with EtOAc (1 L × 3) and dried. The residue (1.6 g) was separated by silica gel (Chromatorex FL100D; Fuji Silysia Chemical, Kozoji-cho, Kasugai, Aichi, Japan) column chromatography (*n*-hexane/EtOAc = 8/2 and 6/4 and CHCl_3_/CH_3_OH = 50/1, 25/1, 13/1, 9/1, 6/4, and 0/1). The CHCl_3_/CH_3_OH = 50/1 fraction was separated by ODS (octadecylsilane) column chromatography (10% to 100% CH_3_OHaq gradient system). The collected fraction (59.3 mg) containing FKJ-0213A was purified by reversed-phase preparative HPLC using a Triart-PFP column (φ20 × 250 mm; YMC, Kyoto, Japan) with a solvent system of 85% CH_3_OH aq containing 0.1% formic acid to yield hatsusamide A (**1**) (11.6 mg). Another collected fraction (65.1 mg) was purified by reversed-phase preparative HPLC using a Triart-PFP column (φ20 × 250 mm; YMC, Kyoto, Japan) with a solvent system of 85% CH_3_OH aq containing 0.1% formic acid to yield tanzawaic acid B (**3**) (1.4 mg). The CHCl_3_/CH_3_OH = 25/1 fraction was separated by ODS column chromatography (10% to 100% CH_3_OH aq gradient system) to yield trichodermamide C (**4**) (28.3 mg).

The EtOAc extract (6.0 g) obtained by the procedure described above from the other-day-cultured broth (6 L) was separated by silica gel column chromatography using the same solvent system. The CHCl_3_/CH_3_OH = 9/1 fraction was separated by ODS column chromatography (10% to 100% CH_3_OHaq gradient system). The collected fraction (59.3 mg) containing hatsusamide B was purified by reversed-phase preparative HPLC using a Triart-PFP (pentafluorophenyl) column with a solvent system of 45% CH_3_OH aq containing 0.1% formic acid to yield hatsusamide B (**2**) (28.3 mg).

Hatsusamide A (**1**): yellow amorphous solid; [α]${}_{D}^{23}$ = +5.4 (*c* = 0.1, CH_3_OH); IR (KBr) ν_max_ cm^−1^ 3289, 2919, 1741, 1653, 1428, 1213; UV (CH_3_OH) λ_max_ (log ε) 332 (4.25), 270 (4.52), 206 (4.67); the ^1^H- and ^13^C-NMR data in (CD_3_)_2_CO are shown in Table 1.

Hatsusamide B (**2**): yellow amorphous solid; [α]${}_{D}^{25}$ = +112.6 (*c* = 0.1, CH_3_OH); IR (KBr) ν_max_ cm^−1^ 3364, 1684, 1617, 1301, 1098; UV (CH_3_OH) λ_max_ (log ε) 326 (4.30), 252 (sh), 229 (sh), 206 (4.43); the ^1^H- and ^13^C-NMR data in CD_3_OD are shown in Table 1.

Tanzawaic acid B (**3**): colorless crystal; HR-ESIMS *m*/*z* 275.2010 [M + H]^+^ (calcd. for C_18_H_26_O_2_, 275.2005); [α]${}_{D}^{25}$ = +44.6 (*c* = 0.1, CH_3_OH); ^1^H-NMR signals (CDCl_3_, 500 MHz) *δ*: 7.38 (dd, *J* = 15.4, 11.0 Hz, 1H), 6.29 (dd, *J* = 15.0, 10.3 Hz, 1H), 6.12 (dd, *J* = 15.0, 11.0 Hz, 1H), 5.77 (d, *J* = 15.4 Hz, 1H), 5.55 (dddd, *J* = 4.6, 4.6, 4.6, 2.9 Hz, 1H), 5.43 (ddd, *J* = 9.5, 1.9, 1.9 Hz, 1H), 2.42 (ddd, *J* = 10.3, 10.3, 5.5 Hz, 1H), 2.18 (m, 1H), 1.80 (m, 1H), 1.71 (dddd, *J* = 12.6, 2.9, 2.9, 2.9 Hz, 1H), 1.63 (ddd, *J* = 13.5, 5.7, 3.7 Hz, 1H), 1.46–1.55 (m, 1H), 1.31–1.40 (m, 1H), 0.94 (d, *J* = 7.4 Hz, 3H), 0.92 (m, 1H), 0.89 (d, *J* = 6.3 Hz, 3H), 0.87 (d, *J* = 6.3 Hz, 3H), 0.71–0.81 (m, 2H).

Trichodermamide C (**4**): yellow amorphous solid; HR-ESIMS *m*/*z* 447.1409 [M + H]^+^ (calcd. for C_21_H_22_N_2_O_9_, 447.1398); [α]${}_{D}^{25}$ = +32.4 (*c* = 0.1, CH_3_OH);^1^H-NMR signals (DMSO-*d*_6_, 500 MHz) *δ*: 8.05 (s, 1H), 7.41 (d, 8.6 Hz, 1H), 7.17 (d, 8.6 Hz, 1H), 5.42 (d, 10.3 Hz, 1H), 5.36 (br d, 10.3 Hz, 1H), 4.12 (br s, 1H), 3.91 (s, 3H), 3.82 (s, 3H), 3.75 (d, 7.2 Hz, 1H), 3.68 (s, 1H), 3.22 (s, 3H), 2.39 (br s, 1H), 2.01 (d, 17.8 Hz, 1H). These two known compounds were identified by comparison with the ^1^H-NMR data and optical rotation values in the literature [12,13,14].

### 3.4. Hydrolysis of ***1***

A sample of **1** (10.3 mg) was dissolved in 2 M NaOH and stirred at room temperature for 2 h. After stirring, the solution was neutralized with 10% H_2_SO_4_ and evaporated. The residue was dissolved in CH_3_OH and separated by preparative thin layer chromatography (TLC) to yield two crude fractions containing tanzawaic acid B and trichodermamide C derived from hatsusamide A. The fraction containing tanzawaic acid B was purified by reversed-phase preparative HPLC using a Triart-PFP column with a solvent system of 85% CH_3_OHaq containing 0.1% formic acid to yield tanzawaic acid B (0.6 mg) derived from hatsusamide A. The fraction containing trichodermamide C was purified by reversed-phase preparative HPLC using a Triart-PFP column with a solvent system of 45% CH_3_OHaq containing 0.1% formic acid to yield trichodermamide C (0.5 mg) derived from hatsusamide A.

### 3.5. X-ray Diffraction Analysis of ***3***

Block crystals of **1** were obtained from methanol solution at room temperature. All measurements were made on a Rigaku-AXIS-Rapid II diffractometer (Matsubara-cho, Akishima, Tokyo, Japan) using graphite monochromated Cu–Ka radiation. The structure was solved by direct methods (SHELXS-97) and refined by full-matrix least-squares on F^2^. All non-hydrogen atoms were refined anisotropically. Hydrogen atoms were placed in ideal positions and refined using the riding model. Detailed crystallographic information for **3** is provided in the Supporting Information.

### 3.6. Cytotoxic Activity of ***1**–**4*** against Several Tumor Cell Lines

Cytotoxic activity of **1**–**4** were measured using WST-8 (Kishida Chemical, Osaka, Japan) against five cell lines, namely A549 (a human lung carcinoma cell line), HeLa S3 (a human cervical cancer cell line), Panc1 (a human pancreas carcinoma cell line), HT29 (a human colon adenocarcinoma cell line), and H1299 (a human non-small lung carcinoma cell line). These cells were gifted from Prof. Shiomi Lab. in Kitasato Institute for Life Sciences. A549, HeLa S3, Panc1, HT29, and H1299 cells were seeded in 96-well plates (5 × 10^3^ cells per well) and cultured in Dulbecco’s modified Eagle medium (Wako Pure Chemical Industries, Osaka, Japan) supplemented with 10% FBS (Fetal Bovine Serum), 100 IU/mL penicillin and 100 μg/mL streptomycin at 37 °C under 5% CO_2_. After culturing the cells overnight, test compounds dissolved in DMSO (Dimethyl Sulfoxide) at appropriate concentrations were added into each well (final concentrations of 1, 3, 10, 30, 100 μM). After 48 h of incubation at 37 °C, WST-8 solution (DOJINDO LABORATORIES, Mashikimachitabaru, Kamimashiki, Kumamoto, Japan) was added to each well, and the cells were further incubated at 37 °C for 4 h. The absorbance at 450 nm of each well was measured using a Corona Grating Microplate Reader SH-9000 (Corona Electric, Ibaraki, Japan). Cell viability was calculated as a percentage relative to the DMSO control.

### 3.7. Anti-Malarial Activity of ***1**–**4***

The evaluation of anti-malarial activity was conducted as previously reported [19]. Briefly, cultured *P. falciparum* (chloroquine-sensitive FCR3 strain and chloroquine-resistant K1 strain) in Type A+ blood was seeded in 96-well culture plates (parasitaemia 0.5–1%, hematocrit 2.0%) and incubated with test compounds (final concentrations of 4.38, 8.75, 17.5, 35, 70 μM) for 72 h in RPMI medium supplemented with 10% human plasma at 37 °C, under 93% N_2_, 4% CO2, and 3% O_2_. After incubation, parasite lactate dehydrogenase activity was assayed to determine the parasite growth and calculate the anti-malarial activity in comparison with the controls that had received no compounds. This study, including the donation of human erythrocytes from volunteers, was approved by the Kitasato Institute Hospital Research Ethics Committee (approval no. 12102).

## 4. Conclusions

In this study, two new compounds **1** and **2** were isolated from the culture broth of the deep-sea-derived fungal strain *Penicillium steckii* FKJ-0213 together with two known compounds, tanzawaic acid B (**3**) and trichodermamide C (**4**). The structure elucidation revealed that **1** was a tanzawaic acid B-trichidermamide C hybrid compound and **2** was a new analog of aspergillazines. Evaluation of the anti-malarial and anti-cancer activities of these four compounds revealed that **1** exhibited anti-malarial and anti-cancer activities and **3** exhibited anti-malarial activity. Interestingly, the constituent compounds of **1**, tanzawaic acid B (**3**) and trichodermamide C (**4**), were individually inactive for anti-tumor activity. In other words, this activity was apparent only after the two compounds were combined. Moreover, the anti-malarial activity of **1** was about three times stronger than that of **3**. These results suggest that the combination of **3** and **4** was responsible for enhancing the anti-tumor and anti-malarial activities. This study provides an interesting example of the identification of a new activity by binding with a known but inactive compound.

## Figures and Tables

**Figure 1 marinedrugs-18-00513-f001:**
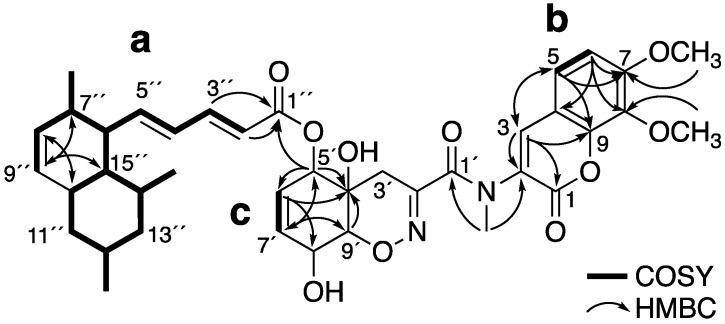
Two-dimensional nuclear magnetic resonance (2D NMR) correlations of **1**.

**Figure 2 marinedrugs-18-00513-f002:**
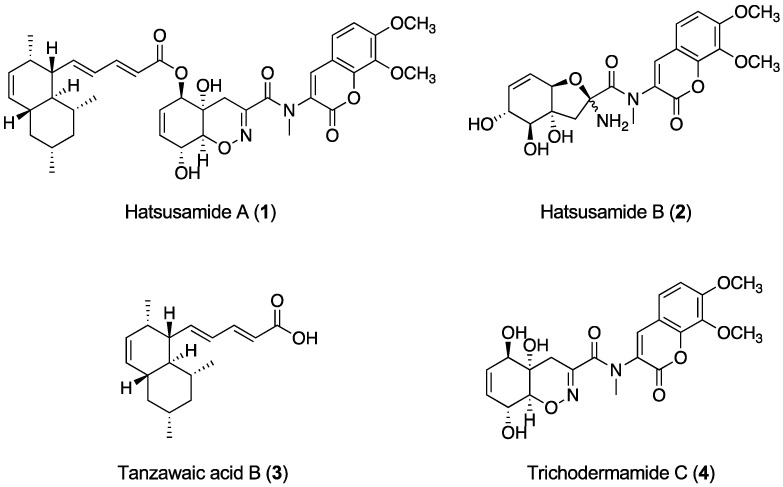
Structures of hatsusamide A (**1**), hatsusamide B (**2**), tanzawaic acid B (**3**), and trichodermamide C (**4**).

**Figure 3 marinedrugs-18-00513-f003:**
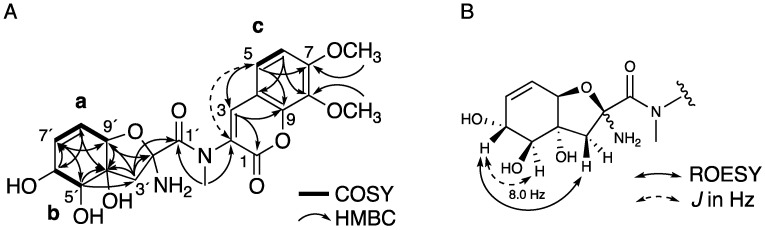
2D NMR correlations of **2**. (**A**) HMBC (Heteronuclear Multiple Bond Correlation) and ^1^H-^1^H COSY (Correlation Spectroscopy) correlations of **2**, (**B**) selected ROESY (rotating-frame nuclear Overhauser effect correlation spectroscopy) correlations and coupling constant of **2**.

**Table 1 marinedrugs-18-00513-t001:** ^1^H and ^13^C 2D nuclear magnetic resonance (NMR) chemical shifts of **1** and **2**.

Hatsusamide A (1) ^a^			Hatsusamide B (2) ^b^	
Position	*δ* _H_	*δ* _C_	*δ* _H_	*δ* _C_
1		159.9		166.8
2		129.6		131.2
3	7.91 (s, 1H)	137.6	7.25 (s, 1H)	119.9
4		114.8		115.5
5	7.39 (d, *J* = 9.2 Hz, 1H)	124.2	6.86 (d, *J* = 8.9 Hz, 1H)	126.4
6	7.13 (d, *J* = 9.2 Hz, 1H)	110.5	6.59 (d, *J* = 8.9 Hz, 1H)	104.5
7		156.4		155.5
8		136.9		137.5
9		147.4		150.9
7-OCH_3_	3.97 (s, 3H)	56.8	3.87 (s, 3H)	56.4
8-OCH_3_	3.90 (s, 3H)	61.3	3.79 (s, 3H)	61.1
N-CH_3_	3.32 (s, 3H)	37.0	2.90 (s, 3H)	35.2
1′		166.3		166.0
2′		152.5		90.7
3′	N.D.	N.D.	2.09 (d, *J* = 14.0 Hz, 1H)	40.6
	N.D.		3.04 (d, *J* = 14.0 Hz, 1H)	
4′		67.4		83.2
5′	5.61 (m, 1H)	76.9	3.70 (d, 8.0 Hz, 1H)	78.8
6′	5.53 (ddd, *J* = 10.4, 2.1, 2.1 Hz, 1H)	126.3	4.14 (m, 1H)	72.1
7′	5.69 (d, *J* = 10.4 Hz, 1H)	131.3	5.62 (ddd, *J* = 10.3, 2.4, 1.4 Hz, 1H)	131.4
8′	4.07 (br s, 1H)	67.9	5.50 (ddd, *J* = 10.3, 3.0, 2.6 Hz, 1H)	127.8
9′	4.01 (br d, 7.4 Hz, 1H)	83.5	4.66 (br dd, 3.0, 2.4 1H)	86.9
1″		167.0		
2″	5.96 (d, *J* = 15.2 Hz, 1H)	119.5		
3″	7.41 (dd, *J* = 15.2, 10.9 Hz, 1H)	146.7		
4″	6.30 (dd, *J* = 14.9, 10.9 Hz, 1H)	127.5		
5″	6.42 (dd, *J* = 14.9, 10.1 Hz, 1H)	151.7		
6″	2.49 (ddd, *J* = 10.1, 9.9, 5.5 Hz, 1H)	50.1		
7″	2.20 (m, 1H)	37.7		
8″	5.61 (m, 1H)	133.0		
9″	5.46 (ddd, *J* = 9.4, 1.9, 1.9 Hz, 1H)	132.9		
10″	1.81–1.88 (m, 1H)	43.4		
11″	0.74–0.83 (overlap, 1H)	42.5		
	1.71–1.77 (m, 1H)			
12″	1.51–1.59 (m, 1H)	33.2		
13″	0.74–0.83 (overlap, 1H)	47.4		
	1.63–1.68 (m, 1H)			
14″	1.37–1.45 (m, 1H)	37.3		
15″	0.97 (overlap, 1H)	47.7		
7″-CH_3_	0.98 (d, 7.4 Hz, 3H)	16.8		
12″-CH_3_	0.88 (d, 6.9 Hz, 3H)	22.7		
14″-CH_3_	0.96 (d, 6.3 Hz, 3H)	23.1		

^a^ Recorded for ^1^H NMR at 500 MHz, ^13^C NMR at 150 MHz in acetone-*d*_6_. ^b^ Recorded for ^1^HNMR at 500 MHz, ^13^C NMR at 150 MHz in methanol-*d*_4_. N.D. = not detectable.

**Table 2 marinedrugs-18-00513-t002:** Anti-malarial activity of **1**–**4** against K1 and FCR3 strains.

Compounds	IC_50_ (μM) Values Against Two Strains	
	K1	FCR3
**1**	27.2	27.9
**2**	>50	>50
**3**	78.5	79.2
**4**	>50	>50
Chloroquine ^a^	0.17	0.038

^a^ Commonly used to treat malaria [17].

**Table 3 marinedrugs-18-00513-t003:** Cytotoxic activity of **1**–**4** against five human tumor cell lines.

Compounds	IC_50_ (μM) Values Against Five Human Tumor Cell Lines				
	HeLa S3	HT29	A549	H1299	Panc1
**1**	15.0	6.8	13.7	18.7	12.9
**2**	>100	>100	>100	>100	>100
**3**	>100	>100	>100	>100	>100
**4**	>100	>100	>100	>100	>100
Staurosporine	<2.0	<2.0	<2.0	<2.0	<2.0

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
