# Peer review of "Hatsusamides A and B: Two New Metabolites Produced by the Deep-Sea-Derived Fungal Strain Penicillium steckii FKJ-0213"

_marinedrugs, 2020, doi:10.3390/md18100513_

Round 1

Reviewer 1 Report

The manuscript by Matsuo et al. describes the isolation and structural characterization of two new metabolites, named hatsusamides A and B, along with two known analogues from the marine fungus Penicillium steckii. The planar structure and configuration of the new compounds were determined by NMR, optical rotation, and chemical hydrolysis analyses. The authors also evaluated anti-malarial and cytotoxic activities of the compounds. Because the new compound (1) is a simple ester of tanzawaic acid B (3) and trichodermamide C (4), and compound (2) is a N-methyl analogue of the known compound aspergillazine D/E, the novelty of this manuscript is minor. Despite, there are some errors and questions on the structural determination, and data presentation in the manuscript needs to be improved.

Major comments:

  1. Page 2, line 66-67. Method used for fungus identification should be cited and include details in the method section.
  2. Page 2, line 75, the molecular formula of compound 1 is wrong. There are ten oxygen atoms, not five.
  3. Page 2, line 76-77. There are two aromatic H and nine olefinic H in compound 1. Same as on page 3, line 115, for compound 2. The current data presentation is very confusing.
  4. Page 3, lines 94-98. The authors indicated that compound 1 was hydrolyzed and analyzed with LC-MS. But there are no related data presented in the main text and supporting info. The authors must include those LC-MS data which clearly show to the readers that compound 1 is indeed an ester of tanzawaic acid B and trichodermamide C. Ideally, a time course of LC-MS analyses with the hydrolyzed crude product can show that 1 degrades into 3 and 4 in a one-to-one ratio. Regarding the comparison of NMR and optical rotation data, proper citations with relevant texts should be included.
  5. Regarding the relative configuration of compound 2, it has been shown that the parental metabolite aspergillazine D/E undergoes the C-2 epimerization (Org Biomol Chem, 2005, 3, 123). How did the authors determine the current configuration of hatsusamides B? Detailed data presentation and discussions are missing.
  6. Some latest papers on the metabolites relevant to the topic should be included, such as J Nat Prod, 2013, 76:1099 and J Nat Prod, 2017, 80: 676.

Author Response

Dear reviewer 1,

Thank you for your kind review. Authors revised the manuscript according to your suggestion. The revision was shown by red color in the revised manuscript.

  1. Page 2, line 66-67. Method used for fungus identification should be cited and include details in the method section.

Our response>

We added the detail method of identification of the strain FKJ-0213 in page 6, line 177-186.

  1. Page 2, line 75, the molecular formula of compound 1 is wrong. There are ten oxygen atoms, not five.

Our response>

Thank you for your attention. We revised it (page 2, line 75).

  1. Page 2, line 76-77. There are two aromatic H and nine olefinic H in compound 1. Same as on page 3, line 115, for compound 2. The current data presentation is very confusing.

Our response>

Thank you for your kind suggestion. We revised it (page 2, line 76 and page 4, line 116).

  1. Page 3, lines 94-98. The authors indicated that compound 1 was hydrolyzed and analyzed with LC-MS. But there are no related data presented in the main text and supporting info. The authors must include those LC-MS data which clearly show to the readers that compound 1 is indeed an ester of tanzawaic acid B and trichodermamide C. Ideally, a time course of LC-MS analyses with the hydrolyzed crude product can show that 1 degrades into 3 and 4 in a one-to-one ratio. Regarding the comparison of NMR and optical rotation data, proper citations with relevant texts should be included.

Our response>

Hydrolysis of 1 inefficiently proceeded because side reaction occurred such as generation of hydrate of trichodermamide C (see the attached file). Thus, we added the figures about comparison data of 1H-NMR of isolated 3 and 4 and hydrolysated 3 and 4 derived from 1 (Figures S7 and S8).

Figure. Total ion current (TIC) chromatogram of hydrolysate of 1.

  1. Regarding the relative configuration of compound 2, it has been shown that the parental metabolite aspergillazine D/E undergoes the C-2 epimerization (Org Biomol Chem, 2005, 3, 123). How did the authors determine the current configuration of hatsusamides B? Detailed data presentation and discussions are missing.

Our response> As you pointed out, we confirmed that the compound 2 probably epimerize by detailed analysis of 1D and 2D NMR of 2. However, in the case of 2, the ratio of diastereomer of 2 is very low (compound 2 : diastereomer = 1 : 0.13) different from aspergillazines D/E. So, we could not assign the 1H NMR of diastereomer. Therefore, the stereochemistry at C-2´ of 2 was revised as shown Figures 2 and 3 in consideration of diastereomer.

  1. Some latest papers on the metabolites relevant to the topic should be included, such as J Nat Prod, 2013, 76:1099 and J Nat Prod, 2017, 80: 676.

Our response> Thank you for your kind information.

Reviewer 2 Report

The article is well written and will be of interest to a certain circle of researchers. The article describes the structural identification of two new compounds. And although the structures are not unique, there are certainly interesting. I have a few comments:

  1. page 2, line 75. the authors made a mistake in the formula of compound 1
  2. In my opinion, the determination of the relative configuration of compound 2 is poorly described. If you take the spectra of compound 2 in DMSO, protons of hydroxyl groups may appear and then you can describe it in more detail.

In addition, you must try to define the absolute configuration for compound 2. There are several options:

  1. to obtain MTPA ethers by the Mosher’s method. There is a high probability that the reaction will take place at the hydroxyl groups in positions 4’ and 6’.
  2. to obtain acetonide at two adjacent hydroxyl groups, if the reaction proceeds through hydroxyls at C-4’ and C-5’, then MTPA ethers can be obtained using the free hydroxyl group at C-6’.

Author Response

Dear reviewer 2,

Thank you for your kind review. Authors revised the manuscript according to your suggestion. The revision was shown by red color in the revised manuscript.

  1. page 2, line 75. the authors made a mistake in the formula of compound 1

Our response>

Thank you for your attention. We revised it (page 2, line 75).

  1. In my opinion, the determination of the relative configuration of compound 2 is poorly described. If you take the spectra of compound 2 in DMSO, protons of hydroxyl groups may appear and then you can describe it in more detail.

Our response>

We believe that the relative configuration of hydroxyl groups of 2 is significantly cleared by ROESY, coupling constants and the background of biogenetic pathway.

In addition, you must try to define the absolute configuration for compound 2. There are several options:

  1. to obtain MTPA ethers by the Mosher’s method. There is a high probability that the reaction will take place at the hydroxyl groups in positions 4’ and 6’.

  1. to obtain acetonide at two adjacent hydroxyl groups, if the reaction proceeds through hydroxyls at C-4’ and C-5’, then MTPA ethers can be obtained using the free hydroxyl group at C-6’.

Our response for the above two comments>

Thank you for your suggestion. However, authors are afraid we do not have sufficient amount of 2 to apply modified Mosher’s method and acetonide.

Reviewer 3 Report

The aim of this manuscript is to characterize and evaluate anti-malarial and cytotoxic activities of two new metabolites produced by the fungal strain Pecicillium steckii. On the whole the manuscript is very well written and the chemical characterization of the two new metabolites has been done well by various suitable techniques. Just minor modifications are needed to improve the quality of the manuscript before publication.

  • In the abstract section the biological activity is presented in µM, therefore, it's important to indicate the molecular weight (MW) of each compound.
  • Specific comment: It is also important to indicate the anti-malarial and cytotoxic activities on the various cell lines of a control compound, in order to compare the biological activity of the two new products to a pure reference product.

Author Response

Dear reviewer 3,

Thank you for your kind review. Authors revised the manuscript according to your suggestion. The revision was shown by red color in the revised manuscript.

  1. In the abstract section the biological activity is presented in µM, therefore, it's important to indicate the molecular weight (MW) of each compound.

Our response>

The molecular weight of each compound was shown in page 7, line 216-234 as accurate mass.

  1. Specific comment: It is also important to indicate the anti-malarial and cytotoxic activities on the various cell lines of a control compound, in order to compare the biological activity of the two new products to a pure reference product.

Our response>

We added the positive controls for anti-malarial (chloroquine) and for anti-tumor (staurosporine) activites and its description in Table 1 and 2 and page 5 lines 152 and 158.

Round 2

Reviewer 1 Report

The authors have made improvements in this revised manuscript and addressed some critical concerns. However, for the purposes of clarity, there are some suggestions to follow up with my prior comments.

Minor comments:

  1. The authors answered prior Q#4 regarding the hydrolyzed product analysis of compound 1 and added comparison data of NMR spectra and attached TIC. As this is important to support the conclusion, I suggest the authors to put the TIC figure in main text or supporting information for scientific dissemination.
  2. The authors responded my Q#5 that “… we confirmed that the compound 2 probably epimerize by detailed analysis of 1D and 2D NMR of 2. However, in the case of 2, the ratio of diastereomer of 2 is very low (compound 2 : diastereomer = 1 : 0.13) different from aspergillazines D/E. So, we could not assign the 1H NMR of diastereomer.” Because the epimerization is important to be informed to the readers in term of structural elucidation, the authors should incorporate their answers in the main text as well. Also, it is worthy of discussions on why the ratio of diastereomer of 2 is low that is different from aspergillazines D/E? I reasonable explanation should be provided.

Author Response

Thank you for your kind suggestions. We revised our manuscript according to your suggestions.

Our comments for Reviewer 1

  1. The authors answered prior Q#4 regarding the hydrolyzed product analysis of compound 1 and added comparison data of NMR spectra and attached TIC. As this is important to support the conclusion, I suggest the authors to put the TIC figure in main text or supporting information for scientific dissemination.

Our response>

We added the TIC chromatogram in supporting information (Figure S6).

  1. The authors responded my Q#5 that “… we confirmed that the compound 2 probably epimerize by detailed analysis of 1D and 2D NMR of 2. However, in the case of 2, the ratio of diastereomer of 2 is very low (compound 2 : diastereomer = 1 : 0.13) different from aspergillazines D/E. So, we could not assign the 1H NMR of diastereomer.” Because the epimerization is important to be informed to the readers in term of structural elucidation, the authors should incorporate their answers in the main text as well. Also, it is worthy of discussions on why the ratio of diastereomer of 2 is low that is different from aspergillazines D/E? I reasonable explanation should be provided.

Our response>

We consider that the only difference from aspergillazines D and E, the N-methyl group, is probably involved in suppression of the epimerization at C-2´ position. This description was added page 4, line139-142 with the description about ratio of diastereomer.

Reviewer 2 Report

The authors have made some changes suggested by the reviewers. Despite the absence of an absolute configuration of stereocenters for compound 2, the article can be accepted for publication.

Author Response

Thank you for your review again.